# Dopamine Receptor Activation Modulates the Integrity of the Perisynaptic Extracellular Matrix at Excitatory Synapses

**DOI:** 10.3390/cells9020260

**Published:** 2020-01-21

**Authors:** Jessica Mitlöhner, Rahul Kaushik, Hartmut Niekisch, Armand Blondiaux, Christine E. Gee, Max F. K. Happel, Eckart Gundelfinger, Alexander Dityatev, Renato Frischknecht, Constanze Seidenbecher

**Affiliations:** 1Leibniz Institute for Neurobiology (LIN), Department of Neurochemistry and Molecular Biology, 39118 Magdeburg, Germany; mitloehnerjessica@gmail.com (J.M.); ablondia@lin-magdeburg.de (A.B.); gundelfi@lin-magdeburg.de (E.G.); 2German Center for Neurodegenerative Diseases (DZNE), Molecular Neuroplasticity Group, 39120 Magdeburg, Germany; Rahul.Kaushik@dzne.de; 3Center for Behavioral Brain Sciences (CBBS), 39120 Magdeburg, Germany; 4Leibniz Institute for Neurobiology (LIN), Department of Systems Physiology of Learning, 39118 Magdeburg, Germany; hartmut.niekisch@gmail.com (H.N.); mhappel@lin-magdeburg.de (M.F.K.H.); 5Center for Molecular Neurobiology Hamburg (ZMNH), Institute for Synaptic Physiology, 20251 Hamburg, Germany; christine.gee@zmnh.uni-hamburg.de; 6Otto-von-Guericke University, Medical Faculty, 39120 Magdeburg, Germany

**Keywords:** ADAMTS 4/5, brevican, chondroitin sulfate proteoglycan, D1/D5 receptors, NMDA receptors

## Abstract

In the brain, Hebbian-type and homeostatic forms of plasticity are affected by neuromodulators like dopamine (DA). Modifications of the perisynaptic extracellular matrix (ECM), which control the functions and mobility of synaptic receptors as well as the diffusion of transmitters and neuromodulators in the extracellular space, are crucial for the manifestation of plasticity. Mechanistic links between synaptic activation and ECM modifications are largely unknown. Here, we report that neuromodulation via D1-type DA receptors can induce targeted ECM proteolysis specifically at excitatory synapses of rat cortical neurons via proteases ADAMTS-4 and -5. We showed that receptor activation induces increased proteolysis of brevican (BC) and aggrecan, two major constituents of the adult ECM both in vivo and in vitro. ADAMTS immunoreactivity was detected near synapses, and shRNA-mediated knockdown reduced BC cleavage. We have outlined a molecular scenario of how synaptic activity and neuromodulation are linked to ECM rearrangements via increased cAMP levels, NMDA receptor activation, and intracellular calcium signaling.

## 1. Introduction

Synaptic transmission and plasticity are affected by perisynaptic and extrasynaptic factors including the extracellular matrix (ECM), glia-derived components, and neuromodulators. The neuromodulator dopamine (DA) plays an important role in classical and newly discovered forms of synaptic plasticity, such as neo-Hebbian or spike-timing-dependent plasticity (STDP), and hence is fundamental to various forms of learning [1,2]. Dopaminergic modulation of synapses lasts from milliseconds to hours and comprises such diverse mechanisms as regulation of presynaptic neurotransmitter release, e.g., via control of axon terminal excitability or calcium influx, postsynaptic neurotransmitter detection via regulated receptor insertion, or synaptic integration in networks (summarized in Reference [1]). Ultimately, dopaminergic activation also contributes to structural spine plasticity [3].

Dopaminergic signaling is mediated via five different G protein-coupled receptors which can be assigned to two major subgroups: D1-like and D2-like DA receptors [4,5,6]. Both receptor subgroups have been shown to be coupled to adenylyl cyclase (AC). D1-like receptor activity leads to increased cAMP levels and activation of protein kinase A (PKA) (reviewed in Reference [7]). D1 receptors (D1Rs) are localized both pre- and postsynaptically [8]. D2 (D2R) and D3 receptors have also been found to be expressed both postsynaptically on DA target cells and presynaptically on dopaminergic neurons [8,9]. DA receptors may interact heterologously with other receptor types, such as AMPA-type (AMPARs) and NMDA-type glutamate receptors (NMDARs). This association seems to be important in regulating long-term potentiation (LTP) and working memory [10,11,12]. NMDARs have been reported to form dynamic surface clusters with D1Rs in the vicinity of glutamatergic synapses. Thus, D1Rs may regulate synaptic plasticity by modulating the synaptic localization of NMDARs [11].

Here, we followed the hypothesis that dopaminergic signaling can also affect the integrity of the hyaluronan (HA)-based extracellular matrix (ECM) that surrounds and stabilizes synapses. This type of ECM structures occurs in the brain as highly condensed perineuronal nets (PNNs) or the more diffuse perisynaptic ECM [13]. The perisynaptic ECM forms a meshwork of macromolecules based on HA as a backbone for chondroitin sulfate proteoglycans like brevican (BC), aggrecan (ACAN), versican or neurocan, glycoproteins, and link proteins (for review, see References [14,15]). As shown by us and others, in-vivo-like ECM structures also develop in dissociated neuronal primary cultures [16,17,18,19]. Interestingly, the HA-based neural ECM is formed and remodeled in an activity- and plasticity-dependent manner under in vivo and in vitro conditions [20,21]. Prime candidates for this remodeling are matrix metalloproteases (MMPs) and disintegrin and metalloprotease with thrombospondin motifs (ADAMTS) enzymes [22]. ADAMTS-4 was identified as one of the major proteases processing the proteoglycans ACAN, BC, neurocan, and versican (reviewed in Reference [23]), and is thus a key candidate for ECM remodeling in the brain [24,25]. Controlled cleavage of these proteoglycans results in the generation of defined N- and C-terminal fragments which remain bound to HA-based ECM structures or cell surfaces, respectively (for BC, the fragment sizes are approximately 53 and 80 kDa). ADAMTS enzymes can intrinsically be inhibited by tissue inhibitors of matrix proteases (TIMPs) and activated in response to central nervous system (CNS) injury or disease. However, the molecular pathways mediating neuronal-activity-related control of ADAMTS has remained elusive [26,27].

The extracellular activity of the tissue plasminogen activator (tPA) protease has been shown to be significantly increased in the nucleus accumbens (NAc) of mice after activation of D1-like DA receptors via a PKA-dependent pathway [28]. This enhanced activity is probably associated with an increased release of tPA from vesicles into the extracellular space upon neuronal stimulation, which in turn regulates homeostatic and Hebbian-type synaptic plasticity [28,29,30]. Here, we hypothesized that proteases modifying the HA-based perisynaptic ECM (i.e., ADAMTS-4 and -5) might also be released or activated after stimulation of D1-like DA receptors. Consequently, we investigated whether activation of DA receptors affects perisynaptic ECM integrity via stimulating the release or activation of ECM-modifying ADAMTS proteases. We found DA receptor agonists that increase cleavage of the ECM constituents BC and ACAN in the rat cortex and in primary cortical cultures, identified the responsible matrix metalloproteases, and unraveled the underlying intracellular signaling mechanisms.

## 2. Materials and Methods

### 2.1. Antibodies and Drugs

The following primary antibodies were used: rabbit anti-ADAMTS-4 (Abcam, Cambridge, UK), rabbit anti-ADAMTS-5 (OriGene, Rockland, MD, USA), rabbit anti-aggrecan (Merck Millipore, Burlington, MA, USA), rabbit anti-aggrecan neoepitope (Novus Biologicals, Centennial, CO, USA), guinea pig anti-brevican (Seidenbecher et al., 1995) (custom-made; central region of rat BC), mouse anti-brevican (BD Biosciences, San José, CA, USA), rabbit anti-brevican “neo” (Rb399) (custom-made; neo-epitope CGGQEAVESE) [21,31], affinity-purified rabbit anti-brevican “neo” (Rb399) (custom-made; neo-epitope CGGQEAVESE), rat anti-D1 dopamine receptor (Sigma-Aldrich, St. Louis, MO, USA), rabbit anti-D2 dopamine receptor (Abcam, Cambridge, UK), mouse anti-GAD65 (Abcam, Cambridge, UK), rabbit anti-GAPDH (SYSY, Göttingen, Germany), rabbit anti-GFAP (SYSY, Göttingen, Germany), mouse anti-Homer 1 (SYSY, Göttingen, Germany), mouse anti-MAP2 (Sigma-Aldrich, St. Louis, MO, USA), and mouse anti-PSD95 (NeuroMab, Davis, CA, USA). Secondary antibodies were: Cy™ 3 goat anti-rabbit IgG (H + L) (Dianova, Hamburg, Germany), Alexa Fluor^®^ donkey anti-mouse 568 IgG (H + L) (Invitrogen, Carlsbad, CA, USA), Alexa Fluor^®^ 488 donkey anti-mouse IgG (H + L) (Invitrogen, Carlsbad, CA, USA), Alexa Fluor^®^ 488 donkey anti-rabbit IgG (H + L) (Invitrogen, Carlsbad, CA, USA), Cy™ 3 donkey anti-guinea pig IgG (H + L) (Dianova, Hamburg, Germany), donkey anti-mouse 647 IgG (H + L) (Invitrogen, Carlsbad, CA, USA), Alexa Fluor^®^ 488 donkey anti-rat IgG (H + L) (Invitrogen, Carlsbad, CA, USA), peroxidase-coupled AffiniPure donkey anti-rabbit IgG (H + L) (Jackson ImmunoResearch, Cambridgeshire, UK), and peroxidase-coupled AffiniPure donkey anti-mouse IgG (H + L) (Jackson ImmunoResearch, Cambridgeshire, UK).

Drugs were used in the following concentrations and obtained from the following companies: SKF38393 hydrobromide (for in vivo application 5 mg/kg body weight, Tocris Bioscience, Bristol, UK), SKF81297 hydrobromide (for in vitro application 1 µM, Tocris Bioscience, Bristol, UK), (−)−Quinpirole hydrochloride (1 µM, Tocris Bioscience, Bristol, UK), SCH23390 hydrochloride (10 µM, Abcam, Cambridge, UK), Tetrodotoxin citrate (TTX, 1 µM, Tocris Bioscience, Bristol, UK), AP-5 (50 µM, Tocris Bioscience, Bristol, UK), Ifenprodil (3 µM, Tocris Bioscience, Bristol, UK), cAMPS-Rp triethylammonium salt (15 µM, Tocris Bioscience, Bristol, UK), recombinant human TIMP-3 (7.5 nM, R&D Systems, Minneapolis, MN, USA), Diltiazem hydrochloride (20 µM, Tocris Bioscience, Bristol, UK), KN93 phosphate (2 µM, Tocris Bioscience, Bristol, UK), and Hexa-D-Arginine (0.58 µM, Tocris Bioscience, Bristol, UK).

### 2.2. Primary Neuronal Cultures

Primary cultures of rat frontal cortical neurons were prepared for immunocytochemistry (ICC) and biochemistry as described previously [32]. In brief, cells were plated on poly-D-lysine-coated glass coverslips at a density of 50,000 cells per coverslip (Ø 15 mm) for ICC, and at a density of 500,000 cells per poly-D-lysine-coated well (Ø 35 mm) for biochemical analysis. Cultures were kept in an incubator at 37 °C and 5% CO_2_ for 21 days in vitro (DIV).

### 2.3. Immunocytochemistry

After drug treatment, living neurons were incubated with antibodies against full-length and cleaved BC (Rb399) in culture medium for 20 min at 37 °C, and then fixed with 4% paraformaldehyde (PFA) in phosphate-buffered saline (PBS) for 5 min at room temperature (RT). Permeabilization and blocking were done by 30 min incubation with blocking solution (10% fetal calf serum (FCS) in PBS, 0.1% glycine, 0.1% Triton-X 100) at RT. Cells were incubated with primary antibodies diluted in blocking solution overnight at 4 °C. On the following day, cells were washed three times with PBS for 10 min, stained with fluorescently labeled secondary antibodies for 45 min at RT in the dark, washed three times with PBS for 10 min and mounted with Mowiol (Carl Roth, Karlsruhe, Germany). Preparations were kept at 4 °C until image acquisition. Images were acquired on a Zeiss Axioplan fluorescence microscope (Carl Zeiss, Oberkochen, Germany) and further processed for quantitative analysis with NHI ImageJ 1.51w (US National Institutes of Health, Bethesda, MD, USA). Quantification of perisynaptic full-length BC and Rb399 intensities was done using OpenView Software (OpenView 1.5, Noam Ziv) (described in [33,34]).

### 2.4. Design of shRNA Plasmids

To knock down rat ADAMTS-4 (GeneID: 66015) and ADAMTS-5 (GeneID: 304135), shRNA plasmids were cloned by the insertion of the siRNA sequences (Dharmacon, Lafayette, CO, USA; Horizon Discovery, Waterbeach, UK) (Table 1) targeting the open reading frame of rat ADAMTS-4 and ADAMTS-5 into AAV U6 GFP (Cell Biolabs Inc., San Diego, CA, USA) using BamH1 (New England Biolabs, Frankfurt a.M., Germany) and EcoR1 (New England Biolabs, Frankfurt a.M., Germany) restriction sites.

### 2.5. siRNA Sequences

### 2.6. Generation of Adeno-Associated Viral Particles

Positive clones were sequenced and used for the production of recombinant adeno-associated particles as described previously in Reference [35]. In brief, HEK 293T cells were transfected using calcium phosphate with an equimolar mixture of the shRNA-encoding AAV U6 GFP, pHelper (Cell Biolabs Inc., San Diego, CA, USA) and RapCap DJ plasmids (Cell Biolabs Inc., San Diego, CA, USA). A full 48 h after transfection, cells were lysed using freeze–thaw cycles and treated with benzonase (50 U/mL; Merck Millipore, Burlington, MA, USA) for 1 h at 37 °C. Lysates were centrifuged at 8000× *g* at 4 °C. Afterwards, supernatants were collected and filtered using a 0.2 micron filter. Filtrates were passed through pre-equilibrated HiTrap Heparin HP affinity columns (GE HealthCare, Chicago, IL, USA), followed by washing with Wash Buffer 1 (20 mM Tris, 100 mM NaCl, pH 8.0; sterile filtered). Columns were additionally washed with wash buffer 2 (20 mM Tris, 250 mM NaCl, pH 8.0; sterile filtered). Viral particles were eluted using elution buffer (20 mM Tris, 500 mM NaCl, pH 8.0; sterile filtered). To exchange elution buffer with sterile PBS Amicon Ultra-4 centrifugal filters with 100,000 Da molecular weight cutoff (Merck Millipore, Burlington, MA, USA) were used. Finally, viral particles were filtered through 0.22 µM Nalgene^®^ syringe filter units (sterile, PSE, Sigma-Aldrich, St. Louis, MO, USA), aliquoted, and stored at −80 °C.

### 2.7. Knockdown of ECM-Modifying Proteases Using shRNA

At DIV14, dissociated rat cortical cultures were infected either with shADAMTS-4, shADAMTS-5, or a scramble construct (2.07 × 10^7^ particles/µL). One week later, infected cells (DIV 21) were treated with SKF81297 for 15 min to stimulate D1-like DA receptors. Afterwards, staining was performed as described above. However, cells were only stained for Rb399 and the synaptic marker Homer 1. Analysis and quantification were performed as indicated above. Knockdown efficiency was verified using biochemical analysis and immunocytochemical staining for either ADAMTS-4 or ADAMTS-5.

### 2.8. Optogenetic Modulation of cAMP in Dissociated Cortical Neurons

To stimulate cAMP levels in dissociated rat cortical neurons, cells (DIV 14) were infected with AAV2/7.Syn-bPAC-2A-tdimer. A 500 ms flash of a 455 nm LED (0.9 mW/mm^2^) was applied to infected cultures at DIV 21. Cells were stained for the synaptic marker Homer 1 and Rb399 at different time points. BC cleavage was analyzed at Homer 1-positive synapses as described above.

### 2.9. Cell Lysis

For cell lysis, culture medium was aspirated and cells were washed twice with ice-cold PBS. Afterwards, cells were incubated with lysis buffer (150 mM NaCl, 50 mM Tris/HCl, pH 8, 1% Triton-X 100) containing a protease inhibitor cocktail (Complete ULTRA Tablets, EDTA-free, EASYpack, Roche Diagnostics, Basel, Schweiz) for 5 min on ice. Cells were scraped off, centrifuged at 10,000× *g* at 4 °C for 15 min, and supernatants were prepared for SDS-PAGE.

### 2.10. In Vivo Pharmacology and Subcellular Brain Fractionation

Adult male Wistar rats were injected with either SKF38393 (5 mg/kg body weight, i.p.) or vehicle as described previously [36]. Rats were anesthetized with isoflurane 1 h after injection, followed by decapitation with a guillotine. For further use, the prefrontal cortex (PFC), hippocampus and rest of the brain were dissected and stored at −80 °C, as described in detail elsewhere [37]. Subcellular brain fractionation was performed according to Reference [38]. Synaptosomal fractions were harvested and incubated with Chondroitinase ABC (Sigma-Aldrich, St. Louis, MO, USA) at 37 °C for 30 min.

### 2.11. SDS-PAGE and Western Blot

Samples were prepared for SDS-PAGE by adding 5× SDS loading buffer (250 mM Tris/HCl, pH 8, 50% glycerol, 10% SDS, 0.25% bromphenol blue, 0.5 M DTT) and heating at 95 °C for 10 min. Subsequently, 5–20% Tris-glycine SDS polyacrylamide gels were run under reducing conditions. Transfer onto PVDF membranes (Merck Millipore, Burlington, MA, USA) was performed according to standard protocols. Membranes were blocked with 5% non-fat milk powder in TBS-T (150 mM sodium chloride, 50 mM Tris, 0.1% (*v*/*v*) Tween20, pH 7.6) for 30 min at RT and immunodeveloped by overnight incubation at 4 °C with primary antibodies. After washing three times with TBS-T for 10 min, membranes were incubated with secondary antibodies for 60 min at RT and washed again three times with TBS-T for 10 min. Immunodetection was performed using an ECL Chemocam Imager (INTAS Science Imaging Instruments GmbH, Göttingen, Germany). Protein quantification was performed with NHI ImageJ 1.51w (US National Institutes of Health, Bethesda, MD, USA).

### 2.12. Statistical Analysis

All statistical analyses and graphical representations were performed using GraphPad Prism 5 (version 5.02, GraphPad Software, San Diego, CA, USA) and SigmaPlot (version 13, Systat Software GmbH, Erkrath, Germany). For statistical comparison between two groups, paired- or unpaired-sample t tests were used. Statistical comparison of multiple groups was performed by an analysis of variance (one-way ANOVA). Dunnett’s multiple comparison test was used for post-hoc comparisons. The Pearson coefficient of correlation was used for analysis of the relationship between cleaved BC and Homer 1. Statistical tests are indicated in the figure captions. For immunocytochemical analysis, four independent experiments were performed, two coverslips per condition prepared, and five cells per coverslip imaged and analyzed. Here, the indicated n number in each caption represents the number of analyzed coverslips. In the case of western blot analysis, the indicated n number in each caption represents four independent experiments.

### 2.13. Ethical Statement

All experimental procedures were carried out in accordance with the EU Council Directive 86/609/EEC and were approved and authorized by the local Committee for Ethics and Animal Research (Landesverwaltungsamt Halle, Germany; 42502-2-1466 LIN) in accordance with the international NIH Guidelines for Animals in Research.

## 3. Results

### 3.1. Increased BC Cleavage in Synaptosomes After D1-like DA Receptor Activation In Vivo

It has been shown previously that the performance of Mongolian gerbils in an auditory learning task is modulated via DA, as the D1/D5 DA receptor agonist SKF38393 injected systemically shortly before, shortly after, or 1 day before conditioning improved learning significantly [36]. In mice performing a similar task, we found changes in the expression levels of BC in the auditory cortex [37]. To test if DA may be involved in changed cortical BC levels, we pharmacologically activated DA D1-like receptors using systemic application of the D1/D5 agonist SKF38393 (5 mg/kg body weight, i.p.) in rats and investigated alterations in the ECM of the PFC where D1Rs are highly expressed (reviewed in Reference [7]). In addition to BC, we also included ACAN in our investigations, since this lectican is also present at perisynaptic sites [39] and is expressed in the cortex, where it has been shown to act as a gatekeeper for physiological plasticity [40]. We analyzed total BC and ACAN expression levels in the homogenate and synaptosomal fractions of the PFC. For detection of proteolytically cleaved BC, we used a specific antibody against the newly emerged C-terminus (neo-epitope; Rb399) after cleavage by ADAMTS-4 or ADAMTS-5 [41,42]. This antibody detects ADAMTS-4/5-cleaved fragments with a size of 53 kDa, but not full-length 145 kDa BC [21]. We found that the amount of full-length BC, as well as the amount of cleaved BC, was unaltered in the homogenate fraction of control and SKF38393-treated animals (Figure 1A,D,E). In the synaptosomal fraction, there was a trend towards reduction in the level of full-length BC by ~32% (Figure 1A,D), while the cleaved fragment was significantly upregulated by nearly 50% in SKF38393-treated animals 1 h after injection (Figure 1A,E). Using the neo-epitope antibody Rb399, we were able to confirm the increase in the amount of cleaved BC by around 50% in the synaptosomal fraction, while levels in the homogenate were unaltered (Figure 1B,F). Accordingly, the levels of full-length and cleaved ACAN were unaltered in the homogenate fraction (Figure 1C,G,H). In the synaptosomal fraction of SKF38393-treated animals, the amount of cleaved ACAN was significantly increased and levels of full-length ACAN were unaffected (Figure 1C,G,H). These data indicate that, indeed, activation of D1-like DA receptors in vivo modulates synapse-associated ECM components on a short time scale.

### 3.2. D1 DA Receptors are Prominently Expressed in Rat Dissociated Cortical Cultures

To investigate the underlying molecular mechanism of DA-regulated ECM tailoring, we used dissociated rat cortical primary neurons which express a mature type of ECM [17]. First, we tested which types of DA receptors were expressed in the cultured neurons. For D2R, expression in rat dissociated cortical cultures has been reported previously [43]. To recapitulate this finding and to prove the expression of D1R in our culture system, we performed immunocytochemical staining at DIV 21. Indeed, both types of DA receptors appeared as little puncta along dendrites and in the vicinity of Homer 1-positive excitatory synapses (Appendix A; Appendix A). While only some D1R-positive puncta were found to be present around GAD65-positive synapses (Appendix A), approximately 45% of D1R-positive puncta appeared in close vicinity of Homer 1-positive excitatory synapses (Appendix A). A similar percentage of co-localization was found for D2R-positive puncta (Appendix A).

### 3.3. Stimulation of D1-Like, But Not D2-Like, DA Receptors Augments Perisynaptic BC Cleavage

In a first in vitro approach, we analyzed BC expression and proteolytic cleavage after the stimulation of D1-like and D2-like DA receptors. To this end, we treated dissociated cortical cultures at DIV 21 for 15 min with the D1/D5 DA receptor agonist SKF81297, which is commonly used for in vitro studies [44,45,46,47] or the D2-like DA receptor agonist quinpirole. Subsequently, we performed immunocytochemical staining to quantify the fluorescent signals of total and cleaved BC. As previously reported by us and others, at this time-point, the cultures had reliably developed a HA-based ECM with similar composition as in vivo [17,18,20]. For detection of proteolytically cleaved BC, we used the above described neo-epitope-specific antibody Rb399.

The perisynaptic amount of total and cleaved BC at excitatory and inhibitory synapses was analyzed using Homer 1 and GAD65 as markers (Figure 2A,C). At Homer 1-positive excitatory synapses, the amount of cleaved BC was significantly increased by up to 186% ± 21% (mean ± SEM) after stimulation of D1-like DA receptors, while the amount of total BC was unaltered (Figure 2B). In contrast, at inhibitory synapses the amount of total as well as cleaved BC remained unaltered upon D1-like or D2-like DA receptor activation (Figure 2D) (data for total BC not shown). Analyzing the amount of total and cleaved BC along whole dendrites revealed no significant changes after activation of either D1-like or D2-like DA receptors (Figure 2E,F) (data for total BC not shown). Since we observed an increase in ACAN cleavage in synaptosomes upon D1R stimulation in vivo (Figure 1C,H), we tested whether this also happened in the culture system. Indeed, after D1-like DA receptor stimulation, the amount of perisynaptically cleaved ACAN was doubled in comparison to control conditions. Similarly to BC, D2-like DA receptor activation had no effect on ACAN cleavage at Homer-positive synapses, and the amount of perisynaptic total ACAN was unaltered in both conditions (Figure 2G,H). Based on these findings, we assume that DA signaling via D1-type receptors has the potential to restructure the perisynaptic ECM at excitatory synapses, while D2-like receptor activation did not induce cleavage in the HA-based ECM.

To prove that the augmented BC cleavage was due to activation of D1-like DA receptors, we applied the selective D1-like antagonist SCH23390 (SCH), which counteracted the SKF81297 effect at excitatory synapses (Figure 2I). The amount of cleaved BC was unaltered at inhibitory synapses and along whole dendrites after D1-like DA receptor inhibition (Figure 2J,K). Thus, perisynaptic BC cleavage at excitatory synapses proved, indeed, to be D1-like-DA-receptor-dependent.

### 3.4. ADAMTS-4 and ADAMTS-5 are Essential for SKF81297-Induced BC Cleavage

Based on previous findings, the most promising candidate proteases for cleaving BC are ADAMTS-4 and ADAMTS-5 [31,48,49,50]. Therefore, we hypothesized these ECM-modifying proteases to be activated upon D1-like DA receptor stimulation. To test this, we blocked ADAMTS-4 and -5, which proved to be expressed in cortical cultures (Appendix A), with TIMP3, an efficient endogenous polypeptide inhibitor of these proteases. Indeed, application of TIMP3 20 min prior to D1-like receptor activation abolished the effect on BC cleavage (Figure 3A).

As TIMP3 inhibits both proteases, we used a knockdown approach to differentiate between their effects. We used recombinantly produced AAVs expressing small interfering RNAs to specifically knock down either ADAMTS-4 or -5 in our culture system. First, we tested the knockdown efficiency and specificity of two different shRNA constructs for each protease using immunocytochemical staining and western blot analysis (Table 1) (data not shown). The shRNA constructs shADAMTS-4.2 (shA4) and shADAMTS-5.2 (shA5) showed the most efficient knockdown of the respective protease in neurons (Table 1) (Appendix A). These constructs were used for further experiments. Next, rat dissociated cortical cultures (DIV 14) were infected with the above described shRNA constructs shA4 and shA5 (Table 1). At DIV 21, infected cells were treated with SKF81297 as described above and stained for Homer 1 and Rb399. As a negative control, cells were infected with AAV expressing scramble shRNA (Scr) (Table 1). Treatment with shRNAs shA4 and shA5 led to a strong decrease of cleaved BC under unstimulated conditions and prevented SKF81297-induced increase in BC cleavage around excitatory synapses (Figure 3B). The concurrent knockdown of ADAMTS-4 and -5 revealed the same results as single knockdowns (Figure 3B), potentially because remaining cleaved BC is not acutely processed but stays bound at synapses. These results indicate that ECM-modifying proteases ADAMTS-4 and -5 are active under basal conditions and become more activated after D1-like DA receptor stimulation, being essential for DA-dependent perisynaptic BC cleavage.

### 3.5. Pro-Protein Furin-Like Convertases Are Involved in DA-Dependent BC Cleavage

ECM-modifying proteases like ADAMTS enzymes are expressed by neurons and astrocytes [51]. To become activated, their N-terminal pro-domain has to be cleaved off by pro-protein convertases (PPCs) [23,52]. All members of the ADAMTS family have at least one site (R/KX_n_R/K↓R) for furin-like PPCs. To test the role of this activating step, we treated cultures with an inhibitor of furin-like PPCs, i.e., Hexa-D-Arginine. Inhibition of PPCs activities indeed abolished the SKF81297-induced increase in perisynaptic BC cleavage, suggesting that DA-dependent activation or release of PPCs is upstream of ADAMTS-4/5 activation (Figure 3C).

### 3.6. Network Activity, Activation of Postsynaptic NMDARs, and Opening of L-Type Ca^2+^ Channels Are Essential for D1-like-DA-Receptor-Induced BC Cleavage

To address the question, whether D1-like DA receptor activation is sufficient to induce increased perisynaptic ECM cleavage, we studied the role of general neuronal activity in this process. By blocking voltage-gated sodium channels using TTX to prevent action potentials and thus network activity, the effect of the D1-like DA receptor agonist on BC cleavage was suppressed (Figure 4A). This suggests that neuronal activity is crucial for DA-mediated BC cleavage.

Since we found changes in cleaved BC particularly around excitatory synapses, where NMDARs are located, and as these receptors have been shown to interact functionally with D1-like DA receptors [53,54], we tested for a role of NMDARs in the signaling mechanisms underlying DA-dependent BC cleavage. To address this, we first blocked all subtypes of NMDARs using AP-5 and successfully abolished BC cleavage increase with this treatment (Figure 4B). Since it has been shown that D1Rs are expressed in close vicinity of GluN2B-containing NMDARs in the PFC [55], we blocked GluN2B in the culture system using Ifenprodil, which again abolished the increase in BC cleavage after D1-like receptor stimulation (Figure 4C). In summary, general network activity as well as functional postsynaptic GluN2B-containing NMDA receptors is essential to increased perisynaptic BC cleavage via DA stimulation.

Reported physiological interactions of D1-like DA receptors and L-type voltage-gated calcium channels (VGCC) suggested a possible influence of these channels in the investigated DA-dependent perisynaptic BC cleavage [44,56,57]. When we blocked postsynaptic L-type VGCC with the antagonist diltiazem hydrochloride, stimulation of D1-like DA receptors did not augment perisynaptic BC cleavage any further (Figure 4D).

### 3.7. Elevated Intracellular cAMP Levels Enhance BC Cleavage

D1-like DA receptors are typically coupled to Gα_s/olf_ and cause increased cAMP levels and PKA activity (reviewed in Reference [7]). Therefore, we investigated whether elevation of intracellular cAMP levels in neurons is sufficient to result in enhanced perisynaptic BC cleavage. To do so, we used blue light-inducible AC encoded by AAV2/7.Syn-bPAC-2A-tdimer for optogenetic modulation of cAMP levels in the primary culture system [58]. BC cleavage was analyzed around Homer 1-positive synapses at different time points after blue light illumination of the cells. In as little as 5 min after light stimulation we observed enhanced perisynaptic BC cleavage up to nearly 150% (Figure 4E). This effect strength is comparable to our findings for pharmacologically induced perisynaptic BC cleavage, which was found 15 min after D1-like DA receptor activation. In the optogenetic approach, the rise in intracellular cAMP levels occurred much faster. As shown by Reference [58], cAMP rises within about 1 s and is back to baseline about 1 min after the light turns off in hippocampal neurons. The optogenetic cAMP elevation was sufficient to cause a brief increase in perisynaptic BC cleavage, since 10 min after light stimulation the levels had already returned to baseline (Figure 4E), potentially due to highly active phosphodiesterases which were not blocked in our experiments.

### 3.8. DA-Dependent BC Cleavage Requires Co-Signaling Through PKA and CaMKII

Based upon this reductionistic approach to show the involvement of cAMP, we further characterized the intracellular signaling cascade. Accordingly, we blocked PKA activity using the cell-permeable competitive antagonist cAMPS-Rp interacting with the cAMP binding sites on the regulatory subunits of PKA, and again analyzed the perisynaptic amount of cleaved BC. PKA inhibition indeed abolished the SKF81297-induced increase in perisynaptic cleaved BC, confirming the hypothesis that stimulation of D1-like DA receptors results in activation/release of the extracellular proteases in a PKA-dependent manner (Figure 4F). As D1R signaling has also been reported to be alternatively mediated via protein kinase C (PKC) [44,59] we also tested for the involvement of this kinase using bisindolylmaleimide II (1µM); however, no effect on DA-dependent perisynaptic BC cleavage was observed (data not shown).

As BC cleavage depends on signaling through NMDARs and L-type VGCC activation, indicating increased Ca^2+^ influx into the activated cells, we finally tested the involvement of Ca^2+^/calmodulin-dependent kinase II (CaMKII) as a major downstream effector of Ca^2+^ signaling. Blocking CaMKII using KN93 abolished the SKF81297-induced effect of increased perisynaptic BC cleavage (Figure 4G), suggesting that co-signaling via PKA and CaMKII is required to promote BC proteolysis.

### 3.9. Increased BC Cleavage at Synapses With More Prominent Homer 1 Expression

Since BC cleavage depends on postsynaptic activity, it is plausible to assume that the local perisynaptic concentration of cleaved BC is related to local postsynaptic organization. Hence, we investigated whether there wsa a correlation between intensities of immunofluorescent signals for cleaved BC and Homer 1 at single synapses under basal conditions and after stimulation of DA receptors. Indeed, after log transformation of intensities, there was an obvious correlation between perisynaptic cleaved BC and postsynaptic Homer 1 for most cells in all culture preparations and studied conditions (Appendix A). Consistently, we obtained a coefficient of correlation around 0.3 for all treatments (Appendix A). Thus, the rule “more Homer, more cleaved BC” was highly reproducible. For SKF81297 treatment—characterized by higher values of cleaved BC—the clusters representing all analyzed synapses per cell were located a bit higher and shifted to the right, as compared to other groups. However, as there was no difference between treatments in terms of linear regression parameters, intercept, and slope (Appendix A), the clusters lay on average along the same regression line for all conditions. Thus, the regulations controlling BC cleavage similarly depend on postsynaptic mechanisms both under basal conditions and after pharmacological stimulation of D1-like receptors.

## 4. Discussion

Here, we demonstrated that D1-like DA receptor stimulation leads to a restructuring of the ECM surrounding excitatory synapses of cortical neurons, as measured by increased cleavage of its major components BC and ACAN. This effect is mediated via the PKA/cAMP pathway and requires network activity and NR2B/CaMKII activation, triggering extracellular proteases ADAMTS-4 and -5 (for a working model, see Appendix A). Our findings suggest that the neuromodulator DA is a potential sculptor of the perisynaptic ECM, translating selective dopaminergic signals into structural plasticity of excitatory synapses. These findings pave the way to a deeper mechanistic understanding of the dopaminergic contribution to neuronal and synaptic plasticity via ECM remodeling.

### 4.1. How Do DA and DA Receptors Contribute to Local Plasticity?

DA has been shown to exert important neuromodulatory control over different forms of plasticity like LTP [60], long-term depression (LTD) [61], or STDP [62] (see Reference [1] for summary). In neurons of the PFC, a key structure for abstract memory and decision making, DA has been shown to act both as a tonic background regulator and as a phasically released modulator of LTP or LTD, following inverted U-functions in both paradigms [63]. The authors reported LTP to be facilitated by tonic background DA levels, while LTD appeared to require acutely released DA for its facilitation.

Dopaminergic mechanisms have been shown to be involved in spine restructuring. For instance, DA increase reverts aberrant structural plasticity in the NAc of alcohol-withdrawn rats by renormalizing the density of long thin spines and restoring limbic memory [3]. In a very elegant study, Yagishita et al. (2014) demonstrated that the DA-mediated support of spine enlargement upon stimulation has to occur within a narrow time window after NMDAR activation [64]. This mechanism of DA action involves D1Rs, PKA, NMDAR, and CaMKII, which is in good agreement with our findings presented here, thus suggesting the hypothesis that the D1Rs-induced ECM remodeling observed by us may be involved in catecholaminergic spine morphology regulation. This is also in good agreement with unpublished findings from our group showing that BC cleavage is increased in the cortex of DAT-Cre mice heterozygously expressing Cre recombinase instead of the dopamine transporter (DAT) under the control of the DAT promoter, potentially due to chronically increased tonic DA levels because of a decrease in the maximum DA uptake rate in these animals [65].

In the PFC as well as in cortical cultures, D1Rs are considered the most prominent DA receptor subtypes, but D2Rs have also been found in cortical areas [66] (reviewed in Reference [7]) and in our in vitro culture system. However, D2R agonists had no effect on perisynaptic ECM proteolysis. Still, the possibility exists that D1R/D2R heterodimers are involved. Since there is evidence that these heteromers are preferentially coupled to Gα_q_ G proteins, which activate PLC-IP3-DAG signaling (reviewed in [1]), and we found PKC not to be involved in the D1R-agonist-induced ECM restructuring, it is rather unlikely that D1/D2 heteromers play a crucial role, although at this point we cannot completely rule out this possibility.

Ito and colleagues (2007) identified the D1R-cAMP-PKA pathway as the underlying mechanism for enhanced extracellular tPA activity [28]. Based on our findings, the perisynaptic BC cleavage upon D1-like DA receptor activation also follows this signaling pathway. Furthermore, Iwakura et al. (2011) showed in striatal cultured neurons DA- and D1R-agonist SKF38393-increased shedding and release of the epidermal growth factor (EGF) ectodomain, concomitant with ADAMTS activation and calcium signaling [67]. Along this line, Li et al. (2016) demonstrated in striatal slices and in cultured astrocytes that the D1R agonist SKF81297 mediated MMP activation, potentiating ß-dystroglycan cleavage and leading to stimulated NMDAR calcium currents [47]. Taken together, the signaling pathway identified in this study (Appendix A) could be of general significance for transducing DA signals into structural and functional consequences at synapses.

### 4.2. Induced ECM Cleavage as a Prerequisite for Local Structural and Functional Plasticity?

Restructuring or disintegration of the synaptic ECM has diverse effects on synaptic properties and characteristics, but most of studies have used exogenous glycosidase treatment rather than controlled endogenous proteolysis. Thus, glycosidic cleavage of ECM components has been proven to modulate synaptic short-term plasticity through the exchange of postsynaptic glutamate receptors [18]. Further adding to this picture, synaptic LTP has also been shown to be impaired after hyaluronidase treatment via involvement of VGCC [68]. Pyka et al. (2011) showed that glycosidic cleavage of HA and chondroitin sulfate leads to an increased number of synaptic puncta and reduced postsynaptic responses in cultured hippocampal neurons [69].

Beyond glycosidase treatment, regulated and limited proteolytic cleavage by extracellular proteases like tPA, MMP-9, neurotrypsin, or ADAM10 has also been shown by some studies to affect synaptic function and structure (summarized in [70]). For instance, the extracellular ADAMTS protease *gon-1* was demonstrated to regulate synapse formation during development in *Caenorhabditis elegans* neuromuscular junctions [71].

Thus, the findings presented here point to the possibility that DA-induced cleavage of perisynaptic ECM components via ADAMTS-4 and/or -5 may directly be involved in the morphological restructuring of excitatory synapses. This hypothesis is consistent with studies in BC knockout mice and in rat slices treated with anti-BC antibodies showing reduced synaptic plasticity [72].

### 4.3. Does Protease Activation Occur in a Compartment-Specific Way?

Enzymes of the ADAMTS family are synthesized as pro-enzymes, which are cleaved at their C- or N-termini by furin or furin-like PPCs such as PACE4, resulting in mature and potentially active enzymes [73,74]. ADAMTS-5 has been shown to be exclusively activated extracellularly [75]. However, local specificity of ADAMTS-4 activation is controversial: while one study claimed the activation of ADAMTS-4 in the trans-Golgi network via furin [76], Tortorella and colleagues (2005) demonstrated that at neutral pH, furin is less efficient in activating pro-ADAMTS-4 than PACE4, and at acidic pH PACE4 is the only PPC to efficiently activate ADAMTS-4 [52]. Here, we demonstrated that furin or furin-like PPCs mediate DA-dependent perisynaptic BC cleavage, which does not happen at dendrites or inhibitory synapses, arguing for a locally restricted activation mechanism.

A recent study identified tPA as another potential activator for ADAMTS-4 in spinal cord injury leading to cleavage of inhibitory chondroitin sulfate proteoglycans (CSPGs) [77]. Interestingly, tPA has been shown to display enhanced extracellular activity depending on the activation of postsynaptic D1Rs in the NAc [28], and it can also be considered a plausible candidate for local ADAMTS-4 activation in our study.

### 4.4. A Critical View on the Role of Astrocytes

Astrocytes are additional players in dopaminergic signaling. Culture studies have provided strong evidence that D1Rs and D5 receptors are expressed in rat astrocytes [78,79,80,81]. Interestingly, the D1-like DA receptor agonist SKF81297 increased intracellular cAMP levels in astrocytes [78,82,83]. It is therefore tempting to speculate that both neuronal and astrocytic D1-like DA receptors may contribute to the local increase in extracellular active proteases.

In our shRNA experiments, the knockdown of ADAMTS-4 and -5 was predominantly achieved in neurons, although we observed ADAMTS-4 expression in astrocytes (Appendix A), which was in line with a previous study showing ADAMTS-4 to be expressed and to cleave BC in neuronal as well as in astrocytic cultures [41]. Thus, the remaining BC cleavage activity in our knockdown experiments may have been due to residual enzymatic activity of astrocytic-derived ADAMTS-4. It is of note that astrocytes are also a rich source of BC [51].

### 4.5. Loss of Full-Length Lecticans vs. Occurrence of Fragments at Synapses

Several functional consequences of the observed increase of cleaved lecticans at excitatory synapses are expected: limited disintegration of the perisynaptic ECM will support receptor mobility, volume transmission of neuromodulators, and structural changes in the synaptic architecture, as discussed above. Recent work suggests that BC controls AMPAR and K_v_1 channel clustering on parvalbumin-positive interneurons [84], and our own preliminary data show a regulation of small-conductance Ca^2+^-activated K^+^ (SK2) channels by BC in principal cells [85], raising the possibility that proteolytic degradation may thus locally change excitability and efficacy of synapses. At the same time, the lectican fragments may exert additional signaling functions. The new and emerging concept of matricryptins as biologically active ECM fragments containing a cryptic site normally not exposed in the intact molecule [86], which was developed for non-neuronal tissues, may also apply to the neural ECM. This view is supported by the fact that the N-terminal BC fragment has been shown to bind to HA and fibronectin, leading to enhanced cell adhesion in glioma [87].

The decline in cleaved BC around synapses only 10 min after stimulation of intracellular cAMP levels (Figure 4B) could point to limited retention of the fragment at perisynaptic sites, where it might act as a bioactive molecule transiently inducing downstream signaling cascades.

## 5. Conclusion: Importance of D1R-Modulated ECM Remodeling for Physiological and Pathological Plasticity

Homeostatic and Hebbian plasticity mechanisms interact to keep neural circuits healthy and adaptable. We have shown that induction of homeostatic plasticity in neuronal cell cultures also increased BC processing at both excitatory and inhibitory synapses [21], arguing for a type-specific plasticity signature of local ECM disintegration.

Under pathological conditions, disturbance of plasticity often goes along with disturbances or malformation of the neural ECM (for review, see [88,89]). These pathological alterations might be related to changes in dopaminergic signaling, as evidenced, for instance, by the deposition of CSPGs in the Lewy bodies in Parkinson patients suffering from a loss of DA neurons [90]. This is in line with a more recent proteomic study showing massive upregulation of the hyaluronan and proteoglycan binding link protein 2 (HAPLN2) in the substantia nigra of these patients [91].

BC and its cleavage fragments are important not only for healthy brain function but also in malignant glioma, the most common and deadly primary brain tumor [92]. Upregulation of BC and its predominantly ADAMTS-derived fragments has been shown to promote glioma invasion, while uncleavable BC is unable to enhance the invasive potential and tumor progression [50]. Evidence for a link between BC proteolysis and synapses comes from the kainate model of epilepsy, where elevation in this ADAMTS-cleaved BC fragment has been shown [31]. The researchers observed that BC proteolysis in the dentate gyrus outer molecular layer is associated with diminished synaptic density, and thus it is plausible that it is involved in the kainate-induced synaptic loss and/or reorganization. Moreover, BC has been shown to be implicated in drug addiction and craving. In heroin-addicted rats, a clear decrease of full length BC in synaptic protein fractions of the PFC has been noted [93], and Lubbers et al. (2016) demonstrated that heterozygous BC mice expressing approximately 50% of the BC levels of wild-type mice showed a progressive increase in cocaine-associated context preference over time, which may also involve an interplay between ECM and dopaminergic modulation [94]. Altogether, these findings highlight that fine-tuned and locally controlled proteolytic processing of mature ECM components may also have important functional implications for the diseased or addicted brain, potentially making it an interesting target for new treatment strategies.

## Figures and Tables

**Figure 1 cells-09-00260-f001:**
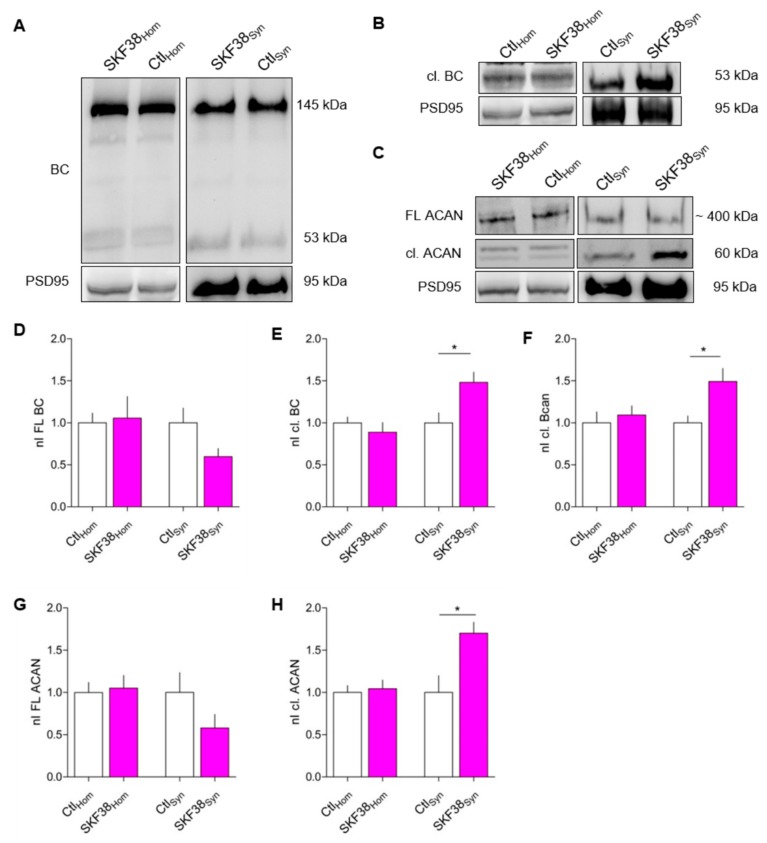
The ECM is altered in synaptosomal fractions of rat prefrontal cortex (PFC) after systemic activation of D1-like dopamine (DA) receptors in vivo. (**A**) Representative western blots showing the expression of brevican (BC) in homogenate (left) and synaptosomal fraction (right) with and w/o SKF38393 treatment. PSD95 illustrates the efficient enrichment of synaptosomes. (**B**) Representative Western blots for the expression of cleaved BC (cl. BC) using the neo-epitope antibody Rb399 in the homogenate (left) and the synaptosomal fraction (right). PSD95 served as an indicator for efficient fractionation. (**C**) Representative western blots for the expression of aggrecan (ACAN) and the cleaved fragment at a size of 60 kDa in the homogenate (left) and the synaptosomal fraction (right). PSD95 served as an indicator for efficient fractionation. (**D**) The levels of full-length BC (FL BC) were unaltered in the homogenate of control and SKF-injected rats. In the synaptosomal fraction, FL BC was reduced after SKF38393 injection (homogenate: Ctl, 1 ± 0.1157, *n* = 4; SKF38, 1.056 ± 0.2574, *n* = 4; average ± SEM; unpaired t test; *p* = 0.8491; synaptosomes: Ctl, 1 ± 0.1745, *n* = 4; SKF38, 0.5980 ± 0.0961, *n* = 4; average ± SEM; unpaired t test; *p* = 0.0901) (nI FL BC = normalized intensity of full-length brevican). (**E**) The expression of cl. BC was unaltered in the homogenate after D1-like DA receptor activation, while in the synaptosomal fraction, the levels of cl. BC were significantly increased after SKF38393 treatment (homogenate: Ctl, 1 ± 0.0696, *n* = 4; SKF38, 0.8875 ± 0.116, *n* = 4; average ± SEM; unpaired t test; *p* = 0.4373; synaptosomes: Ctl, 1 ± 0.12, *n* = 4; SKF38, 1.481 ± 0.1221, *n* = 4; average ± SEM; unpaired t test; * *p* = 0.0308) (nI cl. BC = normalized intensity of cleaved brevican). (**F**) The expression of cl. BC as measured by the antibody Rb399 was unaltered in the homogenate, but significantly increased in the synaptosomal fraction of SKF38393-injected rats (homogenate: Ctl, 1 ± 0.1301, *n* = 4; SKF38, 1.092 ± 0.1098, *n* = 4; average ± SEM; unpaired t test; *p* = 0.8014; synaptosomes: Ctl, 1 ± 0.0826, *n* = 4; SKF38, 1.491 ± 0.1563, *n* = 4; average ± SEM; unpaired t test; * *p* = 0.032) (nI cl. BC = normalized intensity of cleaved brevican). (**G**) Full-length aggrecan (FL ACAN) was unaltered in the homogenate, while it was slightly but not significantly decreased in the synaptosomal fraction after D1-like DA receptor activation (homogenate: Ctl, 1 ± 0.1176, *n* = 4; SKF38, 1.05 ± 0.1511, *n* = 4; average ± SEM; unpaired t test; *p* = 0.8014; synaptosomes: Ctl, 1 ± 0.2342, *n* = 4; SKF38, 0.5798 ± 0.1598, *n* = 4; average ± SEM; unpaired t test; *p* = 0.1889) (nI FL Aggr = normalized intensity of full-length aggrecan). (**H**) The expression of cleaved ACAN (cl. ACAN) was unaltered in the homogenate, while in the synaptosomal fraction the levels were significantly increased after D1-like DA receptor activation (homogenate: Ctl, 1 ± 0.0815, *n* = 4; SKF38, 1.044 ± 0.1017, *n* = 4; average ± SEM; unpaired t test; *p* = 0.7458; synaptosomes: Ctl, 1 ± 0.1985, *n* = 4; SKF38, 1.7 ± 0.1317, *n* = 4; average ± SEM; unpaired t test; * *p* = 0.0261) (nI cl. Aggr = normalized intensity of cleaved aggrecan).

**Figure 2 cells-09-00260-f002:**
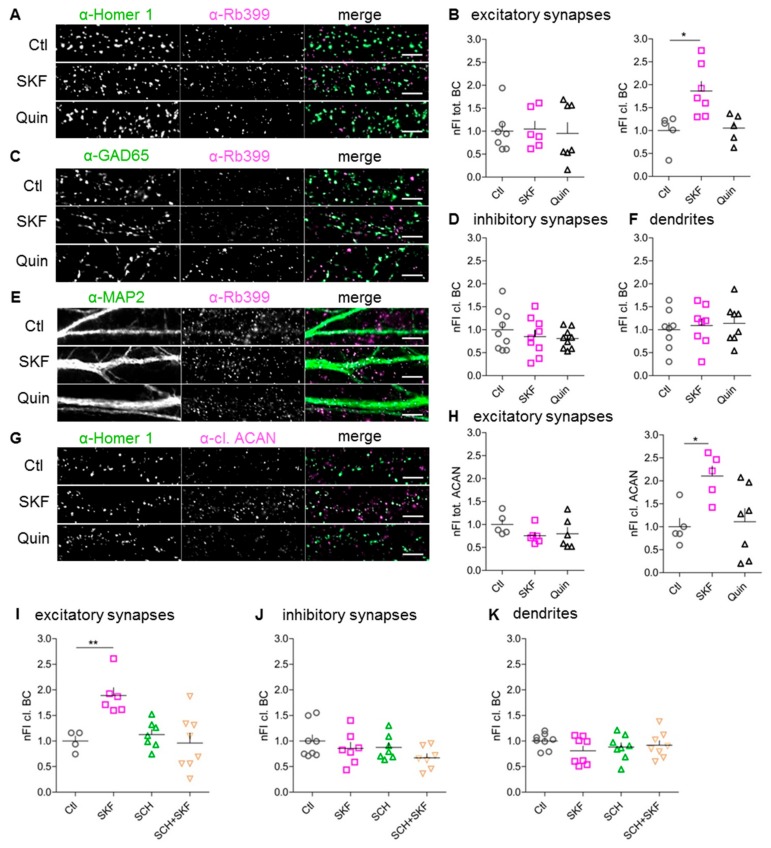
Modification of perisynaptic ECM at excitatory synapses is D1-like-DA-receptor-dependent. (**A**,**C**,**E**) Rat dissociated cortical neurons (DIV21) were either not treated (Ctl) or treated with SKF81297 (SKF) or quinpirole (Quin) and stained for the excitatory synaptic marker Homer 1 (**A**, green), the inhibitory synaptic marker GAD65 (**C**, green) or the somato-dendritic marker MAP2 (**E**, green) and extracellular cleaved BC Rb399 (magenta) (scale bar: 5 µM). (**B**) At Homer 1-positive excitatory synapses, the amount of cleaved BC is significantly increased after D1-like, but not after D2-like DA receptor activation (tot.BC: Ctl, 1 ± 0.1742, *n* = 7; SKF, 1.045 ± 0.1745, *n* = 6; Quin, 0.9502 ± 0.2380, *n* = 7; average ± SEM; one-way ANOVA; *p* = 0.9471; cl. BC: Ctl, 1 ± 0.1673, *n* = 5; SKF, 1.864 ± 0.21, *n* = 7; Quin, 1.054 ± 0.1403, *n* = 5; average ± SEM; one-way ANOVA; *p* = 0.007; Dunnett’s multiple comparison test; * *p* < 0.05). (**D**,**F**) The amount of cleaved BC remained unaltered at inhibitory synapses (**D**) and on dendrites (**F**) after DA receptor activation (**D**): Ctl, 1 ± 0.149, *n* = 9; SKF, 0.8521 ± 0.1359, *n* = 9; Quin, 0.8077 ± 0.0715, *n* = 9; average ± SEM; one-way ANOVA; *p* = 0.5239; (E): Ctl, 1 ± 0.1511, *n* = 8; SKF, 1.09 ± 0.1542, *n* = 8; Quin, 1.139 ± 0.1514, *n* = 8; average ± SEM; one-way ANOVA; *p* = 0.809). (**G**) Rat dissociated cortical neurons (DIV21) were either not treated (Ctl) or treated with SKF or Quin and stained for the excitatory synaptic marker Homer 1 (green) and extracellular cleaved ACAN (cl. ACAN) (magenta) (scale bar: 5 µM). (**H**) ACAN cleavage was also increased at Homer 1-positive excitatory synapses after D1-like, but not D2-like DA receptor activation. The amount of total ACAN was unaltered (tot. ACAN: Ctl, 1 ± 0.1055, *n* = 5; SKF, 0.7565 ± 0.0728, *n* = 6; Quin, 0.7995 ± 0.0728, *n* = 6; average ± SEM; one-way ANOVA; *p* = 0.2938; cl. ACAN: Ctl, 1 ± 0.1859, *n* = 5; SKF, 2.105 ± 0.2179, *n* = 5; Quin, 1.108 ± 0.291, *n* = 7; average ± SEM; one-way ANOVA; *p* = 0.0204; Dunnett’s multiple comparison test; * *p* < 0.05). (**I**) Treatment with SCH23390 (SCH) abolished the SKF81297 (SKF)-induced increase in perisynaptically cleaved BC at excitatory synapses (Ctl, 1 ± 0.0999, *n* = 4; SKF, 1.888 ± 0.1542, *n* = 6; SCH, 1.125 ± 0.0998, *n* = 7; SKF + SCH, 0.9609 ± 0.1892, *n* = 8; average ± SEM; one-way ANOVA; *p* = 0.0014; Dunnett’s multiple comparison test; ** *p* < 0.01). (**J**,**K**) The amount of cleaved BC was unaltered at inhibitory synapses (**B**) and on dendrites (**C**) after D1-like receptor inhibition with SCH (**B**): Ctl, 1 ± 0.1217, *n* = 8; SKF, 0.8552 ± 0.1203, *n* = 7; SCH, 0.8754 ± 0.0924, *n* = 7; SCH + SKF, 0.6704 ± 0.0823, *n* = 7; average ± SEM; one-way ANOVA; *p* = 0.2052; (**C**): Ctl, 1 ± 0.054, *n* = 8; SKF, 0.8099 ± 0.0934, *n* = 8; SCH, 0.883 ± 0.0848, *n* = 8; SCH + SKF, 0.919 ± 0.0893, *n* = 8; average ± SEM; one-way ANOVA; *p* = 0.4384). (nFI tot. BC = normalized fluorescent intensity of total brevican; nFI cl. BC = normalized fluorescent intensity of cleaved brevican; nFI tot. ACAN = normalized fluorescent intensity of total aggrecan; nFI cl. ACAN = normalized fluorescent intensity of cleaved aggrecan).

**Figure 3 cells-09-00260-f003:**
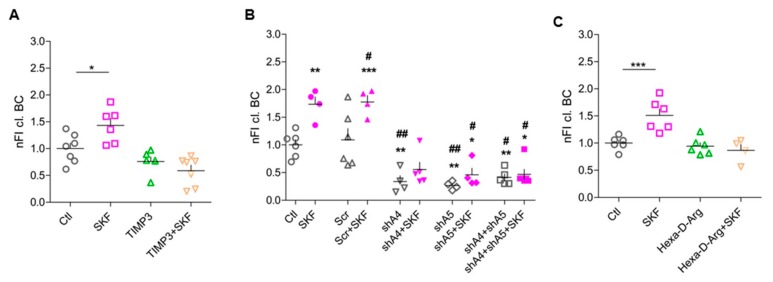
ADAMTS-4 and ADAMTS-5 are essential for D1-like-DA-receptor-induced BC cleavage. (**A**) Inhibition of ADAMTS-4 and ADAMTS-5 with TIMP3 resulted in a decrease in BC cleavage around excitatory synapses (Ctl, 1 ± 0.1013, *n* = 7; SKF, 1.623 ± 0.2566, *n* = 6; TIMP3, 0.7602 ± 0.0851, *n* = 6; SKF + TIMP3, 0.5882 ± 0.1017, *n* = 7; average ± SEM; one-way ANOVA; *p* = 0.0489; Dunnett’s multiple comparison test; * *p* < 0.05). (**B**) Knockdown of ADAMTS-4, ADAMTS-5, or both proteases together led to a significant decrease in D1-like-DA-receptor-induced BC cleavage (Ctl, 1 ± 0.0929, *n* = 6; SKF, 1.732 ± 0.134, *n* = 4; Scr, 1.091 ± 0.2033, *n* = 6; Scr + SKF, 1.773 ± 0.1169, *n* = 4; shA4, 0.3385 ± 0.1052, *n* = 4; shA4 + SKF, 0.5532 ± 0.1343, *n* = 5; shA5, 0.2658 ± 0.0362, *n* = 4; shA5 + SKF, 0.4606 ± 0.1183, *n* = 4; shA4 + shA5, 0.4122 ± 0.0619, *n* = 5; shA4 + shA5 + SKF, 0.4680 ± 0.0911, *n* = 6; average ± SEM; one-way ANOVA; P < 0.001; Dunnett’s multiple comparison test; *** *p* < 0.001) (* = significance compared to Ctl; # = significance compared to Scr). (**C**) ECM-modifying proteases were expressed in an inactive form carrying a pro-domain. Inhibition of the pro-protein convertase PACE4 diminished SKF81927-induced BC cleavage at excitatory synapses (Ctl, 1 ± 0.051, *n* = 6; SKF, 1.509 ± 0.1183, *n* = 6; Hexa-D-Arg, 0.9441 ± 0.0636, *n* = 6; Hexa-D-Arg + SKF, 0.8651 ± 0.1077, *n* = 4; average ± SEM; one-way ANOVA; *p* = 0.0002; Dunnett’s multiple comparison test; *** *p* < 0.001). (nFI cl. BC = normalized fluorescent intensity of cleaved brevican).

**Figure 4 cells-09-00260-f004:**
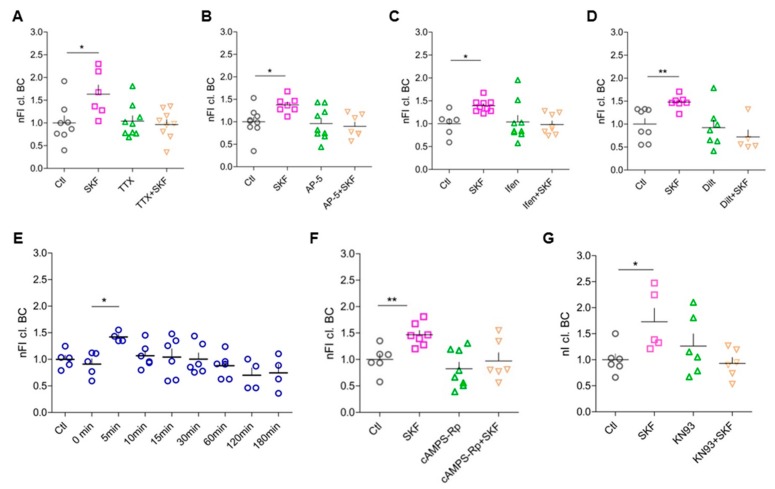
SKF81297-induced BC cleavage requires entire network activity, activity of postsynaptic sites, and co-signaling through PKA and CaMKII.(**A**) Silencing neuronal networks with TTX resulted in an unaltered BC cleavage (Ctl, 1 ± 0.1599, *n* = 8; SKF, 1.632 ± 0.2036, *n* = 6; TTX, 1.039 ± 0.1211, *n* = 9; SKF + TTX, 0.9663 ± 0.1061, *n* = 9; average ± SEM; one-way ANOVA; *p* = 0.0168; Dunnett’s multiple comparison test; * *p* < 0.05). (**B**) Inhibition of all types of NMDARs with AP-5 abolished SKF81297-induced perisynaptic BC cleavage (Ctl, 1 ± 0.105, *n* = 9; SKF, 1.376 ± 0.0671, *n* = 7; AP-5, 0.9599 ± 0.1183, *n* = 9; SKF + AP-5, 0.8976 ± 0.0964, *n* = 7; average ± SEM; one-way ANOVA; *p* = 0.0187; Dunnett’s multiple comparison test; * *p* < 0.05). (**C**) In particular, inhibition of NR2B-containing NMDARs with Ifenprodil (Ifen) resulted in unaltered BC cleavage (Ctl, 1 ± 0.107, *n* = 6; SKF, 1.392 ± 0.046, *n* = 9; Ifen, 1.039 ± 0.1436, *n* = 9; SKF + Ifen, 0.984 ± 0.0782, *n* = 8; average ± SEM; one-way ANOVA; *p* = 0.0199; Dunnett’s multiple comparison test; * *p* < 0.05). (**D**) SKF81297-induced perisynaptic BC cleavage depended on signaling via L-type VGCC (Ctl, 1 ± 0.1223, *n* = 8; SKF, 1.478 ± 0.0542, *n* = 7; Dilt, 0.9228 ± 0.1698, *n* = 7; Dilt + SKF, 0.7188 ± 0.1541, *n* = 5; average ± SEM; one-way ANOVA; *p* = 0.0046; Dunnett’s multiple comparison test; ** *p* < 0.01). (**E**) Elevated intracellular cAMP levels were crucial for an increase in BC cleavage around excitatory synapses (Ctl, 1 ± 0.0765, *n* = 5; 0 min, 0.9096 ± 0.1017, *n* = 7; 5 min, 1.42 ± 0.0481, *n* = 5; 10 min, 1.067 ± 0.0943, *n* = 6; 15 min, 1.042 ± 0.1552, *n* = 6; 30 min, 1.003 ± 0.1172, *n* = 6; 60 min, 0.8787 ± 0.0938, *n* = 6; 120 min, 0.6993 ± 0.1398, *n* = 4; 180 min, 0.7450 ± 0.1605, *n* = 4; average ± SEM; one-way ANOVA; *p* = 0.0207; Dunnett’s multiple comparison test; * *p* < 0.05). (**F**) Increased perisynaptic BC cleavage after D1-like DA receptor stimulation depended on PKA activation (Ctl, 1 ± 0.1044, *n* = 6; SKF, 1.468 ± 0.0805, *n* = 7; cAMPS-Rp, 0.8243 ± 0.1249, *n* = 8; cAMPs-Rp + SKF, 0.9714 ± 0.1567, *n* = 6; average ± SEM; one-way ANOVA; *p* = 0.0041; Dunnett’s multiple comparison test; ** *p* < 0.01). (**G**) SKF81297-induced perisynaptic BC cleavage was subjected to intracellular Ca^2+^ signaling (Ctl, 1 ± 0.1139, *n* = 6; SKF, 1.648 ± 0.2978, *n* = 5; KN93, 0.9165 ± 0.1118, *n* = 4; SKF + KN93, 0.9303 ± 0.1126, *n* = 6; average ± SEM; one-way ANOVA; *p* = 0.0269; Dunnett’s multiple comparison test; * *p* < 0.05). (nFI cl. BC = normalized fluorescent intensity of cleaved brevican.)

**Table 1 cells-09-00260-t001:** siRNA sequences.

**Rat ADAMTS 4**	*shRNA 1: 5′- ATCGTGACCACATCGCTGT -3′*
	*shRNA 2: 5′- TATAGCGCAAGCTGACTGC -3′*
**Rat ADAMTS 5**	*shRNA 1: 5′- TAGCGCGCATGCTTGACTG -3′*
	*shRNA 2: 5′- ATCCCCGTAAACTCGTTCG -3′*
**Control shRNA (Scramble)**	*scrRNA: 5′- CGGCTGAAACAAGAGTTGG -3′*

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
