# Peer review of "Dopamine Receptor Activation Modulates the Integrity of the Perisynaptic Extracellular Matrix at Excitatory Synapses"

_cells, 2020, doi:10.3390/cells9020260_

Round 1
Reviewer 1 Report
In the manuscript entitled “Dopamine modulates the integrity of the perisynaptic 3 extracellular matrix at excitatory synapses”, Mitlöhner et al. presents the molecular pathway of how D1-type dopamine (DA) receptors are linked to perisynaptic extracellular matrix (ECM) modifications. The authors found that activation of D1-like DA receptors (but not D2-like receptors) increased brevican (BC, a major component of ECM) cleavage. They also identified the underlying proteases, ADAMTS-4 and ADAMTS-5. In addition, they reported key molecules involved in the proposed pathways, including pro-protein furin-like convertases, NMDARs, L-type Ca2+ channels, cAMP, PKA, CaMKII, and Homer 1. I found the manuscript interesting and scientifically sound. I recommend the authors to consider my following suggestions.
Major:
1. Dopamine and dopamine receptors are distinct molecules. To my understanding, the authors used drugs that act on dopamine receptors to test the proposed pathways, but did not experiment on dopamine. I wonder if it is possible for the authors to test the pathways by directly experimenting on dopamine. If not, then it is safer to put only dopamine receptors (but not dopamine) in the pathway. This concern is particularly necessary considering that D2-like receptors have no involvement in the pathway. Moreover, is it possible for the authors to use alternative approaches (except using drugs) to manipulate the dopamine receptors?
2. It may be beneficial to add more introduction on BC and its cleavage. For example, it will be difficult for readers to interpret Figure 1A without knowing the molecular weights of the full-length BC and the cleaved BC.
Minor:
The writing of the manuscript can be improved:
The punctuation “Figure 1.” “Figure 2:” should be consistent.
Supplementary Figure S3 is out of order!
Supplementary Figure S6, explain the difference between solid lines and dashed lines in the pathway.
Author Response
Dopamine and dopamine receptors are distinct molecules. To my understanding, the authors used drugs that act on dopamine receptors to test the proposed pathways, but did not experiment on dopamine. I wonder if it is possible for the authors to test the pathways by directly experimenting on dopamine. If not, then it is safer to put only dopamine receptors (but not dopamine) in the pathway. This concern is particularly necessary considering that D2-like receptors have no involvement in the pathway. Moreover, is it possible for the authors to use alternative approaches (except using drugs) to manipulate the dopamine receptors?
The reviewer is right: in this manuscript we just show the effect of 2 different D1R agonists, but not an effect of dopamine itself. We therefore changed the title of our manuscript from "Dopamine modulates..." to "Dopamine receptor activation modulates...". Furthermore, we adjusted two section subheadings in the discussion part to be more precise in our conclusions (lines 542, 719). Finally, in the schematic illustration of the discussed pathway we mark SKF-induced activation of D1 receptors (Suppl. Figure S5).
Indeed, we tried alternative approaches and measured higher Brevican cleavage also in heterozygous DAT-Cre mice showing higher dopamine levels, indicating physiological relevance of our pharmacological study (discussed in lines 556 to 560).
2. It may be beneficial to add more introduction on BC and its cleavage. For example, it will be difficult for readers to interpret Figure 1A without knowing the molecular weights of the full-length BC and the cleaved BC.
We agree with the reviewer and now include more information in the introduction (lines 79 to 81) and in the results (lane 245).
Minor:
The writing of the manuscript can be improved:
The punctuation “Figure 1.” “Figure 2:” should be consistent.
We fixed this.
Supplementary Figure S3 is out of order!
We are sorry for this mistake and fixed it.
Supplementary Figure S6, explain the difference between solid lines and dashed lines in the pathway.
We provide now an explanation in the last lane of the fugure caption.
Reviewer 2 Report
This manuscript written by Mitlöhner et al report an increase in the cleavage of ECM constituents at excitatory synapses following stimulation of D1-like DA receptors. The authors have identified responsible matrix metalloproteases and intracellular pathways mediating this effect. The experiments are nicely designed and performed, the results are clearly described and illustrated.
However, there appears to be quite a mess in the supplemental figures, as the order of the numbers is not right (passing from S2 to S4 directly, Fig S3 is at the end), and the referencing in the manuscript is wrong (particularly for S3, S4 and S5). In addition, the title of Fig S2 does not seem right as it is mentioning D1R, but the content is only about D2R. All the supplemental section should be revised concerning the order, the relation between titles and contents, and the referencing in the manuscript.
Referencing of a result figure in line 313 is also incorrect.
The manuscript appears otherwise well written and documented.
Author Response
However, there appears to be quite a mess in the supplemental figures, as the order of the numbers is not right (passing from S2 to S4 directly, Fig S3 is at the end), and the referencing in the manuscript is wrong (particularly for S3, S4 and S5). In addition, the title of Fig S2 does not seem right as it is mentioning D1R, but the content is only about D2R. All the supplemental section should be revised concerning the order, the relation between titles and contents, and the referencing in the manuscript.
->We apologize for the poor presentation of our supplementary figures and fixed the problem now.
Referencing of a result figure in line 313 is also incorrect.
->We corrected this mistake.